# Integrated miRNA and mRNA Sequencing Reveals the Sterility Mechanism in Hybrid Yellow Catfish Resulting from *Pelteobagrus fulvidraco* (♀) × *Pelteobagrus vachelli* (♂)

**DOI:** 10.3390/ani14111586

**Published:** 2024-05-27

**Authors:** Shu Li, Qiao Yang, Maohua Li, Yue Lan, Zhaobin Song

**Affiliations:** 1Sichuan Key Laboratory of Conservation Biology on Endangered Wildlife, College of Life Sciences, Sichuan University, Chengdu 610065, China; lishu5112@163.com (S.L.); qiao.yang@ki.se (Q.Y.); limaohuahu@foxmail.com (M.L.); 17844626668@163.com (Y.L.); 2Key Laboratory of Bio-Resources and Eco-Environment of Ministry of Education, College of Life Sciences, Sichuan University, Chengdu 610065, China

**Keywords:** hybrid yellow catfish, germplasm characteristics, gonadal development, sterility mechanism, transcriptome

## Abstract

**Simple Summary:**

Although the prevalence of hybrid sterility in fish is a serious limitation to the quality development of the hybrid fish farming industry, it could be useful for the conservation of the genetic stock of endangered and valuable species. In this study, a total of 1709 DEGs were identified by performing RNA sequencing between hybrid and two pure yellow catfishes. A KEGG pathway analysis indicated that some genes related to reproductive functions were upregulated, including genes involved in the cell cycle and oocyte meiosis, whereas genes related to ECM–receptor interactions were downregulated. In total, 63 DEmiRNAs were obtained between the three species of yellow catfishes. The upregulated DEmiRNAs *ipu-miR-194a* and *ipu-miR-499* were found to target the spermatogenesis-related genes *CFAP70* and *RSPH6A*, respectively. Both acted as negative regulators, and this may underscore the miRNA–mRNA regulatory mechanism of sterility in hybrid yellow catfish (*P. fulvidraco* ♀ × *P. vachelli* ♂).

**Abstract:**

The hybrid yellow catfish exhibits advantages over pure yellow catfish in terms of fast growth, fast development, a high feeding rate, and strong immunity; additionally, it is almost sterile, thus ensuring the conservation of the genetic stock of fish populations. To investigate the sterility mechanism in hybrid yellow catfish (*P. fulvidraco* (♀) × *P. vachelli* (♂)), the mRNA and miRNA of the gonads of *P. fulvidraco*, *P. vachelli*, and a hybrid yellow catfish were analyzed to characterize the differentially expressed genes; this was carried out to help establish gene expression datasets to assist in the further determination of the mechanisms of genetic sterility in hybrid yellow catfish. In total, 1709 DEGs were identified between the hybrid and two pure yellow catfishes. A KEGG pathway analysis indicated that several genes related to reproductive functions were upregulated, including those involved in the cell cycle, progesterone-mediated oocyte maturation, and oocyte meiosis, and genes associated with ECM–receptor interaction were downregulated. The spermatogenesis-related GO genes *CFAP70*, *RSPH6A*, and *TSGA10* were identified as being downregulated DEGs in the hybrid yellow catfish. Sixty-three DEmiRNAs were identified between the hybrid and the two pure yellow catfish species. The upregulated DEmiRNAs *ipu-miR-194a* and *ipu-miR-499* were found to target the spermatogenesis-related genes *CFAP70* and *RSPH6A*, respectively, playing a negative regulatory role, which may underscore the miRNA–mRNA regulatory mechanism of sterility in hybrid yellow catfish. The differential expression of *ipu-miR-196d*, *ipu-miR-125b*, and *ipu-miR-150* and their target genes *spidr*, *cep85*, and *kcnn4*, implicated in reproductive processes, was verified via qRT-PCR, consistent with the transcriptome sequencing expression trends. This study provides deep insights into the mechanism of hybrid sterility in vertebrate groups, thereby contributing to achieving a better understanding and management of fish conservation related to hybrid sterility.

## 1. Introduction

The yellow catfish (*Pelteobagrus fulvidraco*), belonging to Bagridae, is a teleost fish found in East and South Asia. It has become one of the most important freshwater aquaculture species in these regions due to its excellent meat quality [1,2]. It is also a benthic omnivorous economic fish common in rivers and lakes in China. However, the disease resistance of yellow catfish fry is poor, and bacterial infection causes a high mortality [3]. Additionally, males and females exhibit prominent sexual size dimorphism (SSD), with adult males growing faster and two- to three-fold bigger than adult females [4,5]. In addition to *P. fulvidraco*, *Pelteobagrus vachelli*, which also belongs to Bagridae, is an important commercial omnivorous species in China [6]. Unlike *P. fulvidraco*, *P. vachelli* has the advantages of strong disease resistance, a low feed coefficient, and a large adult size [5]. However, both species of yellow catfish exhibit considerable size differences among commercially grown fish due to SSD and overwintering behavior, and this significantly affects breeding management and reduces economic benefits.

The determination, differentiation, and developmental regulation of sex in fish are crucial biological processes related to various aspects such as genetic inheritance, breed improvement, and population expansion. Moreover, the development of their gonads directly affects their aquaculture efficiency [7]. When a purebred germplasm is limited in production, crossbreeding is often used to obtain new excellent characters [8]. The hybrid yellow catfish “Huangyou 1” (GS-02-001-2018), resulting from the hybridization of *P. fulvidraco* ♀ (female) × *P. vachelli* ♂ (male), has become a popular breeding trend [5,9]. The hybrid yellow catfish offers advantages over pure yellow catfishes in terms of fast growth and development, a high feeding rate, and strong immunity [10]. Hybrid yellow catfish are almost sterile, which seriously limits the development of the hybrid yellow catfish aquaculture industry; however, hybrid sterility helps the conservation of the genetic stock of yellow catfish [11,12].

Hybrid sterility has always been a troubling problem, such as in rice [13,14], maize [15], wheat [16], cattle-yak [17], drosophila [18], and fish [19,20], and it also greatly limits the development of economic species. Various reasons for hybrid sterility have been studied, such as abnormal meiosis, chromosomal aberrations, and structural variation in particular trait loci [21,22,23]. Additionally, genes play important roles in hybrid sterility, as reported frequently in animals and plants [24,25]. Hybrid yellow catfishes exhibit hybrid sterility, which manifests as reduced biological fitness due to reproductive abnormalities or infertility, leading to limitations in ecological performance [26]. Hybrids often exhibit unfavorable traits due to disruptions in gene expression regulation or disruptions in nuclear–mitochondrial gene interactions [27]. Heterosis, however, cannot be inherited by the next generation, limiting the sustainable utilization of heterosis.

In the past, there were few studies on the mechanism of the sterility of hybrid yellow catfish. Only one anatomical and cytological study focused on the gonadal development and fertility potential of hybrid yellow catfish [28], and it revealed that, during the sexual maturation period, most of the nutrients in hybrid yellow catfish are directed towards body growth, resulting in significant body growth advantages, but ultimately leading to the defective development of both male and female gonads. Specifically, hybrid yellow catfish exhibit degenerated white linear tissue in the ovaries with no ovum production, and the testes show a cavity-like structure with a significantly lower total sperm count and motility than purebred yellow catfish, ultimately leading to hybrid infertility. Currently, there is insufficient research to explain the molecular mechanisms underlying the infertility of hybrid yellow catfish. An examination of the morphology and histology of the gonads in sterile hybrid fish has revealed various impacts on sexual maturation, influenced by the specific combination of parental species. Certain hybrid fish exhibit gonads with a normal size and structure; however, they produce gametes that are morphologically and/or karyotypically abnormal, or gametes that are fertilizable but ultimately non-viable. Further studies are needed to explore the life processes and regulatory mechanisms controlling sterility in hybrid yellow catfish.

MicroRNAs (miRNAs) are small non-coding single-stranded RNAs, approximately 22 nucleotides in length. MicroRNAs (miRNAs) play important roles in the post-transcriptional regulation of gene expression by targeting mRNAs [29,30,31]. Numerous studies have shown that miRNAs can affect many biological processes, such as growth, proliferation, and apoptosis [32,33,34]. Transcriptomic studies on the gonads of the genus *Pelteobagrus* have revealed that genes such as *dmrt1*, *amh*, and *wt1* play pivotal roles in sex determination and differentiation [31,35]. In hybrid yellow catfish, researchers have focused on the effects of heterosis [9], the regulatory mechanisms of a high-fat diet [36], and the regulation of *miR-489-3p* on the oxidative stress response [37] by analyzing mRNA–miRNA in the liver tissue of *P. fulvidraco*, *P. vachelli*, and hybrid yellow catfish. However, to date, no investigation has focused on the mRNA–miRNA regulatory mechanism related to sterility in hybrid yellow catfish.

In this study, we sequenced the gonads of three yellow catfish, *P. fulvidraco*, *P. vachelli*, and hybrid yellow catfish, to reveal the gene expression differences among them and to explore the growth advantage of and sterility mechanism in hybrid yellow catfish, using both mRNA-seq and miRNA-seq. Additionally, the integrity of the internal organs and functional differentiation of yellow catfish have greater similarity to higher vertebrates and humans [38,39]. Therefore, we expected to identify the original mechanism of hybrid sterility in lower vertebrates, providing references for subsequent precision breeding and targeted hybridization, as well as helping to achieve a better understanding of the genetic conservation of species.

## 2. Materials and Methods

### 2.1. Sample Collection and Ethics Statements

The samples of ovaries and testes used in this study were collected from three sexually mature females and three males of *P. fulvidraco*, three sexually mature females and three males of *P. vachelli*, and six hybrid yellow catfish resulting from the cross of a female *P. fulvidraco* and a male *P. vachelli*. All fishes used were over 2 years old, ensuring gonad maturation. All measured samples were collected from the breeding farm of Chuangfu Fisheries Co., Ltd. in Dongpo District, Meishan City, China. One week of temporary rearing was conducted before the experiment. During the temporary rearing period, each recirculating tank was filled with 0.6 m^3^ of water; the pH value of the water body was maintained at 7.6–8.0, the temperature of the water was controlled at 20 °C ± 0.5 °C, and the dissolved oxygen was maintained at 7.5 mg/L or above. Feed was added three times a day (in 6 h intervals), and the daily feeding amount was 3% of the fish mass. The water in the culture tanks was aerated, 20–25% of the water was replaced every day, and the daily photoperiod ratio within the experimental environment was 12 h:12 h. All fishes were euthanized using 2-phenoxyethanol (Sigma, Shanghai Sigma High-Tech Co., Ltd., Shanghai, China) before being dissected. Testes and ovaries were removed surgically, quickly frozen in liquid nitrogen, and then transferred to the laboratory to be stored at −80 °C. All procedures involving animals were approved by the Animal Ethics Committee of the College of Life Sciences at Sichuan University (SCU240508001), and they were conducted in strict adherence to the animal welfare and research laws in China.

### 2.2. Extraction of Total RNA and Transcriptome Sequencing

The total RNA of the sample was extracted using TRIzol reagent (Invitrogen, Carlsbad, CA, USA) following the manufacturer’s instructions. The quantity and purity of the total RNA were assessed using 2% agarose gel electrophoresis and an Agilent 2100 (Santa Clara, CA, USA). Finally, the total RNA purity with OD260/280 was assayed using a Nanodrop, and the total RNA concentration was accurately quantified using Qubit 2.0 (Thermo Fisher, Shanghai Aybio Co., Ltd., Shanghai, China).

After the above operation, rRNA was removed from the total RNA samples using a Ribo-ZeroTM kit (Epicentre, Epocentre Biotechnology Co., Ltd., Madison, WI, USA). First, fragmentation buffer was added to the enriched RNA to break it into small fragments. Then, the first strand of cDNA was synthesized via reverse transcription with 6 bp random hexamers, and the second strand was synthesized by adding buffer, dNTPs (dTTP in dNTP was replaced by dUTP), DNA polymerase I, and RNase H. After purification, terminal repair, the addition of A, and the ligation of the sequencing adaptor, the second strand containing U was degraded by a user enzyme and enriched via PCR. Finally, the PCR product was purified using AMpure XP beads to obtain the final strand specific library. After constructing the library, preliminary quantification was carried out using Qubit 2.0, followed by the subsequent dilution of the library. Then, the Agilent 2100 was used to detect the insert size of the library. When the insert met the expectation, the RT-qPCR method was used to accurately quantify the effective concentration of the library to ensure the quality of the library. According to the requirements of effective concentration, different libraries were pooled into FlowCell, and the Illumina high-throughput sequencing platform was used after clustering Cbot, which generated approximately 150 bp paired-end raw reads.

In order to ensure the accuracy of subsequent analyses, NGSQC Toolkit version 2.3.3 software [40] was used to remove joint sequences, empty sequences, and low-quality sequences (including sequences with N bases > 10% and Q-value < 20) in order to obtain clean reads. A reference genome index was established using Bowtie2 (Version 2.5.2), and then Tophat2 (Version 2.1.1) was used to align with the reference genome of yellow catfish. The reference genomic data were retrieved from the NCBI database for yellow catfish (GCF_003724035.1). Featurecounts software (Version 2.0.2) [41] was used to count the number of reads compared to the genome and obtain the expression matrix. A differential expression analysis was performed using DESeq2 (Version R 3.4.4) [42], with criteria set to FDR < 0.05 and fold change > 1 in order to identify differentially expressed genes (DEGs).

### 2.3. Small-RNA Library Sequencing and Analysis

Total RNA was extracted from gonad tissues using a Small RNA Sample Pre-Kit. RNA adaptors were ligated to the 3′ and 5′ ends of the total RNAs, and the adaptor-ligated sRNAs were used for cDNA synthesis. The target DNA fragments were separated using PAGE gel electrophoresis. After the construction of the library, Qubit 2.0 was used for preliminary quantification, and the library was diluted to 1 ng/μL. Then, the Agilent 2100 was used to check the insert size of the library. When the insert size met the expectation, the Q-PCR method was used to accurately quantify the effective concentration of the library (>2 nM) to ensure the quality of the library. Then, the different libraries were pooled and sequenced on an Illumina HiSeq 2500-SE50 platform (JiTai, Shanghai JiTai Biotechnology Co., Ltd., Shanghai, China) in line with the manufacturer’s instructions.

The raw sequences underwent further processing, including the removal of adaptor reads and the elimination of low-quality reads meeting specific criteria: reads with an N_percent > 10%, those with a base quality ≤ 5 exceeding 50%, and those either exceeding 50% with lengths > 30 nt or <18 nt. Sequences falling within the range of 18~30 bp were selected based on typical miRNA lengths. The resulting clean reads were compared with those in the Rfam database (version 11.0, http://rfam.xfam.org/, accessed on 1 September 2020) and NCBI (http://www.ncbi.nlm.nih.gov/, accessed on 10 September 2020) using blastn (NCBI). Reads mapped to the yellow catfish genome were then analyzed to remove rRNA, tRNA, snoRNA, ncRNA, etc., using RepeatMasker. Meanwhile, the miRNA sequences of 16 bony fish species were downloaded as the background set in miRbase (http://www.mirbase.org/, accessed on 5 April 2021). Among them, *Ictalurus punctatus* was selected as the reference species due to its close relationship with yellow catfish, while the other 15 bony fish species were *Astatotilapia burtoni*, *Cyprinus carpio*, *Danio reio*, *Electrophorus electrocus*, *Fugu rubripes*, *Gadus morhua*, *Hippoglossus hippoglossus*, *Metriaclima zebra*, *Neolamprologus brichardi*, *Oryzias latipes*, *Oreochromis niloticus*, *Pundamilia nyererei*, *Paralichthys olivaceus*, *Salmo salar*, and *Tetraodon nigroviridis*.

Finally, known and novel miRNAs were identified from the remaining sequences using miRDeep2 software (Version 2.0.1.2) [43].

### 2.4. GO, KEGG, and HP Enrichment Analyses

We performed GO category, KEGG pathway, and HP enrichment analyses using g:Profiler (https://biit.cs.ut.ee/gprofiler/gost, accessed on 25 April 2021) [41]. The species that we chose when enriched was Channel catfish (*Ietalurus punetaus*), because there was no yellow catfish in g:Profiler. For the significance threshold, we chose Benjamini–Hochberg FDR < 0.05.

### 2.5. Prediction of microRNA Target Genes and Differential Expression of Known miRNAs

miRanda software (Version 1.0) was used to predict miRNA target genes [44]; the thresholds were set to scores ≥ 155 and free energy of ΔG ≤ −20 kcal/mol. miRNA mainly targets the 3′UTR of the gene, and a length of 500 bp was used as the 3′UTR after the terminator of the gene was extracted. 

The quantifier.pl in miRDeep2 software was employed to obtain the miRNA expressions in the gonad tissues. The differentially expressed miRNAs (DEmiRNAs) between the samples were identified using the edgeR package [45]. miRNA expression was calculated as counts per million (CPM) with edgeR software (Version 1.20.0). The miRNAs with a *p*-value < 0.05 and |log2 fold change| > 1 were selected as DEmiRNAs.

### 2.6. Real-Time Quantitative PCR (qPCR)

Among the DEGs and DEmiRNAs, two candidate genes (*cd79a* and *aldh1a3*) and four candidate miRNAs (*ipu-miR-489*, *ipu-miR-49*, *ipu-miR-194a*, and *ipu-miR-107b*) were randomly selected for RT-qPCR validation to verify the gene expression levels in the yellow catfishes. *β-actin* and *ssc-U6* were used as internal reference genes, in accordance with the previous literature. Fluorescent quantitative primers for candidate genes were designed using the gene coding region (CDS region) sequences from the NCBI online system and Primer 5.0. Finally, the primers were synthesized by Tsingke Biotechnology Co., Ltd. (Beijing, China).

The reaction system was 20 μL, and it contained 10 μL of 2 × T5 Fast qPCR Mix (SYBR Green I), 0.8 μL of upstream and downstream primers, 1 μL of cDNA, and 7.4 μL of DDH2O. The reaction conditions were as follows: 95 °C pre-denaturation (1 min) for one cycle; 95 °C denaturation (15 s); 60 °C annealing (15 s); and 72 °C extending (30 s). The steps of denaturation extended for 40 cycles, and the melting stage for one cycle consisted of 95 °C (5 s), 60 °C (1 min), 95 °C (heating up by 0.11 °C/s), and 50 °C (30 s). A standard curve was constructed using three repeated experimental groups.

### 2.7. Statistical Analysis

After obtaining the CT (threshold cycle) values of the different genes in each sample, the relative expression of the target genes was calculated using Excel (Version 16.17) according to the 2^−ΔΔCT^ method. Then, the two groups of data were tested using the *t* test. A *p*-value ≤ 0.05 was considered to indicate a statistically significant difference. Using GraphPad Prism 8.0 for drawing, the *p*-values were marked as “ns” (>0.05), “*” (≤0.05), “**” (≤0.01), and “***” (≤0.001).

## 3. Results

### 3.1. Data Collection and Transcriptome Assembly

To systematically investigate the transcriptome and expression profile differences among the three yellow catfishes, we collected and sequenced the gonads of the following yellow catfishes: six individuals of the hybrid, six individuals of *P. fulvidraco* (three males and three females), and six individuals of *P. vachelli* (three males and three females). However, one of the male *P. vachelli* was clustered into *P. fulvidraco* in a principal component analysis (PCA) (Appendix A), so it was removed. We finally used two males and three females of *P. vachelli*. All yellow catfish specimens were over two years old to ensure gonad maturation. Detailed sample information is provided in Appendix A. After removing the low-quality reads and adaptors, a total of 79 million 150 bp paired-end clean reads were obtained. Then, all of the clean reads were mapped to the reference genome (ASM372403v1) [4], with an average mapping rate of 66.23% (Appendix A).

### 3.2. Identification of DEGs between Yellow Catfish Species

Next, we processed all of the samples with the same bioinformatics pipeline to identify differentially expressed genes (DEGs) (Table 1 or Figure 1). First, we focused on the DEGs among the hybrid yellow catfish (*P. fulvidraco* (♀) *× P. vachelli* (♂), *P. fulvidraco*, and *P. vachelli*). In total, we identified 1709 DEGs between the hybrid and two pure yellow catfishes, with 1272 upregulated and 437 downregulated DEGs in the hybrid group (Appendix A). Additionally, 2093 genes were identified as DEGs between the hybrid and *P. fulvidraco*, with 1467 upregulated and 626 downregulated DEGs in the hybrid yellow catfishes (Appendix A). Between the hybrid and *P. vachelli*, there were 4694 DEGs, with 4476 upregulated and 218 downregulated in the hybrid yellow catfishes (Appendix A). The upregulated DEGs predominated in all three groups. The difference between the hybrid and *P. vachelli* was greater than the difference between the hybrid and *P. fulvidraco*, with 4094 DEGs, the most abundant (Appendix A). Next, we compared the DEGs across the three groups and found a relatively abundant overlap of DEGs, totaling 780 (Appendix A).

Then, we identified the DEGs between the sexes. The samples of *P. fulvidraco* and *P. vachelli* were mixed together and divided into male and female groups. The hybrid yellow catfish constituted a distinct group due to the ambiguous gonadal distinctions between the sexes. Finally, we identified 3444 DEGs, 1844 upregulated and 1600 downregulated, between the hybrid yellow catfish and the male group (Appendix A). Additionally, 6778 genes were identified as DEGs between the hybrid yellow catfish and the female group, with 6715 upregulated and only 63 downregulated DEGs (Appendix A). There were 814 common DEGs among the three groups of hybrids vs. pure, hybrid vs. male, and hybrid vs. female (Appendix A). An enrichment analyses of the DEGs showed significant development- and sterility-related results.

### 3.3. Enrichment Analysis of DEGs

Then, we performed the Gene Ontology (GO) category, Kyoto Encyclopedia of Genes and Genomes (KEGG) pathway, and Human Phenotype (HP) ontology enrichment analyses to better understand the biological roles of the DEGs.

We used the upregulated and downregulated DEGs between the hybrid and pure yellow catfishes for the enrichment analyses. In the upregulated DEGs of the hybrid yellow catfish, we mainly enriched development-related GO categories, such as system development, animal organ development, and multicellular organism development (Figure 2A and Appendix A). Additionally, the immune system process category was also enriched. Only one KEGG pathway, cytokine–cytokine receptor interaction, was enriched (Appendix A). Some inflammation-related HP categories were enriched, such as inflammatory abnormality of the skin, abnormal inflammatory response, and increased inflammatory response (Appendix A). In the downregulated DEGs of the hybrid yellow catfish, we enriched many cilium-related GO categories, such as cilium assembly, cilium organization, and cilium movement (Figure 2B and Appendix A). Similarly, only one KEGG pathway, purine metabolism, was enriched (Appendix A). In the HP enrichment analysis, we noticed that there were many categories related to sterility, such as abnormal sperm motility, male infertility, decreased fertility in males, and female infertility (Figure 2C and Appendix A).

Similarly, we also performed an enrichment analysis with the DEGs between the hybrid and pure female or male yellow catfish. Firstly, we performed enrichment analyses using the DEGs between the hybrid and pure female groups. We enriched many development-related GO categories, such as developmental process, multicellular organismal process, system development, and tissue development (Appendix A).

Furthermore, only four KEGG pathways were enriched, and no HP category showed enrichment (Appendix A). Then, we performed enrichment analyses using the DEGs between the hybrid and pure male groups. In addition to the enrichment analysis between the hybrid and pure female groups, we also enriched several development-related GO categories between the hybrid and pure male groups, such as developmental process and system development (Appendix A).

However, we enriched other cilium- and reproduction-related GO categories, including cilium assembly, cilium organization, reproduction, and reproductive process (Appendix A). While no KEGG pathway showed significant results, numerous HP categories were enriched, mostly those related to sterility, such as infertility, abnormal sperm motility, ectopic pregnancy, decreased fertility, male infertility, and female infertility (Appendix A).

### 3.4. Identification of miRNAs

We separately constructed small-RNA libraries and sequenced the gonad tissues of the three yellow catfishes to characterize miRNAs. Based on distribution statistics, the lengths of the clean reads showed clear enrichment in two peaks. One of the peaks had a length of 22 nt, the typical size of mature miRNA. According to previous research [46], there are also small RNAs similar to piRNAs in the male testes of vertebrates, with a length of approximately 29 nt (Appendix A).

In total, we obtained 55,147,052 unique reads as candidates to identify miRNA, with an average mapping rate of 74.92% to the yellow catfish genome (ASM372403v1) (Appendix A).

Next, using the known miRNAs from the 16 teleost fishes as the background set, we predicted miRNAs in the yellow catfish (see Materials and Methods). In total, we identified 210 known miRNAs and 156 novel miRNAs across the three yellow catfish species using miRDeep2 (Appendix A). The expression levels of the known miRNAs are shown in Appendix A.

### 3.5. Annotation of the Known miRNAs

To gain a deeper understanding of miRNA function, we predicted the target genes of the known miRNAs and conducted GO and KEGG enrichment analyses on them. In our results, we predicted 15,347 target genes for 210 known miRNAs (see Appendix A). The number of target genes per miRNA ranged from 24 to 3274 across the 210 miRNAs (refer to Appendix A). Our enrichment analyses of target genes revealed a variety of metabolism-related GO categories, including cellular metabolic process, metabolic process, and primary metabolic process (see Appendix A). Additionally, we identified enrichment in six KEGG pathways, with the most significant pathway being aminoacyl-tRNA biosynthesis (also shown in Appendix A). The HP categories primarily pertained to morphological abnormalities (as detailed in Appendix A).

### 3.6. Identification of DEmiRNA and Enrichment Analyses of Target Genes of DEmiRNAs

We identified the differentially expressed miRNAs (DEmiRNAs) between the different groups and performed enrichment analyses on the target genes of these DEmiRNAs. In total, we obtained 63 DEmiRNAs between the hybrid and two pure groups, with 15 up- and 48 downregulated, and we obtained 1747 and 5135 target genes for the upregulated and downregulated DEmiRNAs, respectively (Appendix A). Remarkably, we obtained the same 38 DEmiRNAs (24 upregulated and 14 downregulated) between the hybrid and male groups and between the hybrid and female groups, with consistent upregulation and downregulation directions (Appendix A). Then, we performed enrichment analyses using the target genes of these DEmiRNAs, and both of the groups mentioned above showed significant metabolism-related categories (Appendix A).

### 3.7. Identification of DEmiRNAs and Enrichment Analyses of Target Genes of DEmiRNAs

In addition, we identified overlapping genes between the DEGs and the target genes of 353 DEmiRNAs across the hybrid and two wild groups. We obtained 245 intersection genes between the upregulated DEGs and the target genes of the downregulated DEmiRNAs, and we obtained 42 intersection genes between the downregulated DEGs and the target genes of the upregulated DEmiRNAs (Figure 3 and Appendix A). The expression of these genes may be negatively regulated by miRNAs. We also performed enrichment analyses using these 287 intersection genes, and only some GO biological process and HP categories were enriched (Appendix A). Notably, cilium-related processes, cell migration, and pathways related to intracellular substance synthesis and metabolism were identified, including cilium assembly, cilium organization, RNA biosynthetic processes, and macromolecule metabolic processes.

### 3.8. RT-qPCR Verification

Finally, we performed reverse transcription–quantitative real-time PCR (RT-qPCR) to verify the gene expression levels of two genes and four miRNAs in all samples. Among the DEGs and DEmiRNAs, two candidate genes (*cd79a* and *aldh1a3*) and four candidate miRNAs (*ipu-miR-489*, *ipu-miR-499*, *ipu-miR-194a*, and *ipu-miR-107b*) were randomly selected for RT-qPCR validation. As expected, we found significant increases in the expression of *cd79a*, *aldh1a3*, *ipu-miR-489*, *ipu-miR-499*, and *ipu-miR-194a* and a decrease in the expression of *ipu-miR-107b* in the hybrid and the pure yellow catfishes (Figure 4A,B). The expression changes of the genes between the hybrid and pure yellow catfishes are consistent with the gonad transcriptome results (Figure 4A,B).

## 4. Discussion

### 4.1. DEGs between Hybrid and Pure Yellow Catfishes

The present study utilized the transcriptome sequencing of hybrid yellow catfish, an important economic fish, to discover the mechanism of hybrid sterility in hybrid yellow catfish (*P. fulvidraco* ♀ × *P. vachelli* ♂). Through a comparison of the transcriptomic differences between hybrid and pure male yellow catfish testes, we identified a total of 2756 differentially expressed genes (DEGs), comprising 1526 upregulated genes and 1230 downregulated genes. The genes involved in downregulation were identified as *CFAP70*, *TTC29*, *CCDC40*, *DNAAF4*, *SYCE3*, *STK36*, *SPI1*, and *EMX2*. These genes were found to be enriched in various pathways related to sex development, sperm growth, and sperm motility, such as organelle assembly, cilia movement, the development of the urogenital system, sperm production, and sperm deformity [47,48]. Significantly, various studies have shown that these genes contribute to reproductive development. For example, the gene *CFAP70* (cilia- and flagella-associated protein 70) has been shown to be associated with motile cilia and flagella [49], and its mutations can lead to male infertility [50]. The gene *RSPH6A* has also been reported to be involved in flagellum formation, and the knockout of the gene in mice results in male sterility [51]. Additionally, the deficiency of the *TSGA10* gene can also lead to male infertility in mice [52]. The downregulation of these DEGs in hybrid yellow catfish compared to purebred yellow catfish may explain the mechanism of hybrid sterility in this species. Therefore, we hypothesize that the significant downregulation of DEGs (*CFAP70*, *TTC29*, *CCDC40*, *DNAAF4*, *SYCE3*, *STK36*, *SPI1* and *EMX2*) in hybrid yellow catfish is the primary cause of male infertility or reduced fertility. In addition, in a comparison of the transcriptomic differences between hybrid female and purebred female yellow catfish ovaries, a total of 2353 DEGs were identified, comprising 2328 upregulated genes and 25 downregulated genes. The upregulated genes were found to be enriched in pathways related to organ development, anatomical structure generation, egg incubation and apoptosis, and the formation of epithelial cells and tissues. These pathways significantly affect the migration and differentiation of epithelial cells in the early stage of ovarian development. The representative genes involved in regulating these pathways included *EMX2*, *SPI1*, *FOXD3*, *PDE1A*, *GCM2*, *ERG*, *OSR1*, *RBCK1*, *SMTNL2*, and *MOS*. Therefore, we speculate that the significant upregulation of DEGs (*EMX2*, *SPI1*, *FOXD3*, *PDE1A*, *GCM2*, *ERG*, *OSR1*, *RBCK1*, *SMTNL2*, *MOS*) is the main cause of the lack of ovarian development or reduced fertility in hybrid female yellow catfish.

### 4.2. Sexual Dimorphism in Hybrid Sterility of Yellow Catfish

In this study, we also compared the transcriptomic differences between females and males using the gonads of hybrid yellow catfish. A total of 1085 DEGs were identified, comprising 379 upregulated genes and 706 downregulated genes. The upregulated genes were enriched in three KEGG pathways, mainly involved in the cell cycle, oocyte maturation (map 04914), and meiosis (map 04114), and the key regulatory genes involved in these pathways were *CDC20*, *CDC25B*, *CCNB2*, *MOS*, *CCNB1*, *PLK1*, *CCNA1*, *CCNA2*, *CPEBl*, and *FBXO43*. The downregulated genes were mainly enriched in pathways related to energy binding, cell proliferation, and organ morphogenesis. It is speculated that the upregulated expression of sexually dimorphic genes within the species may promote the development of hybrid yellow catfish towards the female direction, affecting processes such as meiosis, the cell cycle, and oocyte maturation in the gonadal tissue [53]. However, the downregulated expression of sexually dimorphic genes may promote the development of hybrid yellow catfish towards the male direction, affecting processes such as energy metabolism, cell proliferation, adhesion, and morphogenesis in the gonadal tissue [54]. These differences ultimately lead to the occurrence of hybrid infertility in hybrid yellow catfish.

### 4.3. Functional Enrichment of DEGs in Hybrid Yellow Catfishes

In the GO and HP classification, the DEGs between the male and female hybrid yellow catfishes were assigned to three GO terms and four HP terms. The GO and HP annotation results indicate that the fundamental mechanism of female hybrid yellow catfish infertility lies in the abnormal synthesis of progesterone and maturation-promoting factors, leading to the inability of stage II and stage III oocytes to mature and sustain. Additionally, changes occurred in the regulation of transcription, DNA templates (transcription), and the regulation of cellular macromolecule biosynthetic processes in females, resulting in the inability to synthesize a large amount of yolk and nuclear substances. These factors explain the complete infertility of females, as their ovaries mainly remain in stage II without the production of mature oocytes.

Furthermore, in our functional annotation study of the testes of male hybrid yellow catfish, we found that multiple pathways were regulated, such as male infertility, abnormal sperm motility, infertility, abnormality of reproductive system physiology, decreased fertility, abnormal male reproductive system physiology, decreased fertility in males, and functional abnormality of male internal genitalia. These terms revealed the main regulatory mechanisms for the low sperm vitality of male hybrid yellow catfish. Specifically, the synthesis of sperm is downregulated, resulting in the absence or a scarcity of sperm in most hybrid yellow catfish males. In a few individuals that can produce sperm, there are also abnormal regulations in the structure and morphology of sperm flagella, leading to decreased vitality. These findings are consistent with the conclusion of Hu et al., 2019 [28], who reported low sperm vitality and reproductive incapacity in aged hybrid yellow catfish males. Together, these results illustrate that abnormal gonadal development in hybrid yellow catfish is a critical mechanism leading to infertility.

The pathways that were analyzed using the KEGG classification and that appeared to be abnormally expressed in the hybrid yellow catfish were found to have direct links to fertility. Upregulated pathways included the cell cycle, progesterone-mediated oocyte maturation, and oocyte meiosis. We speculate that these may be involved in germ cell development, which may have indirect effects on behavior and the activities and functions of the gonad. The only downregulated pathway was ECM–receptor interaction. ECM is an extracellular structural framework composed of various proteins, including collagen and fibronectin. ECM receptors are membrane proteins on the cell surface, and they can interact with ECM structural proteins. Previous research has shown that the ECM–receptor interaction plays an important regulatory role in follicle maturation, ovulation, and regression [55]. This process involves multiple signaling pathways and molecular mechanisms, which are of great significance for the normal function of the reproductive system.

### 4.4. Expression Validation Using qRT-PCR

Our study also focused on the miRNA regulation of sterility in hybrid yellow catfishes. We identified 210 mature miRNAs in the yellow catfish and 63 DEmiRNAs between the hybrid and two pure yellow catfishes. Our experimental results show that *ipu-miR-196d*, *ipu-miR-125b*, and *ipu-miR-150* and their target genes *spidr*, *cep85*, and *kcnn4* are involved in the developmental regulation of female gonads and their related tissues. They significantly affect biological processes such as ovarian underdevelopment, ovarian apoptosis, and energy allocation in oocyte synthesis. Additionally, *ipu-miR-218a* and its target gene *dpy19l1* are involved in the regulation of male sperm production. The regulation of this microRNA significantly affects male reproductive capacity.

To evaluate the expression profiles of the miRNAs obtained using transcriptome sequencing, seven aberrantly expressed miRNAs were selected for expression validation using a qRT-PCR analysis. We found that the expression levels of the miRNAs in the testes and ovaries of the hybrid yellow catfish were significantly higher than those in the purebred yellow catfish, which indicates that there is a significant involvement of miRNA in the negative feedback regulation of mRNA during the biological process of gonadal development in hybrid yellow catfish. Compared to in the purebred yellow catfish, these miRNAs greatly inhibit mRNA expression, leading to stagnation or poor fertility in gonadal development. Additionally, we also found a significant differential expression of *miR-141-* and *let-7*-family miRNAs in the male reproductive cells of the hybrid yellow catfish. Their higher expression levels significantly inhibit the expression of pathways related to male sperm production and testicular development. These findings are consistent with those of the gonads of fruit flies [56]. Therefore, it can be concluded that *miR-141-* and *let-7*-family miRNAs participate in regulating the biological processes underlying male infertility in hybrid yellow catfish.

## 5. Conclusions

In our research, we identified the DEGs and DEmiRNAs between hybrid and pure yellow catfishes and identified the negative miRNA–mRNA regulatory mechanism of sterility in hybrid yellow catfish. This research could provide a basis for exploring yellow catfish hybrid sterility and improving its economic value. Furthermore, our findings provide a reference for the medical phenomenon or disease of hybrid sterility in higher vertebrates and humans.

## Figures and Tables

**Figure 1 animals-14-01586-f001:**
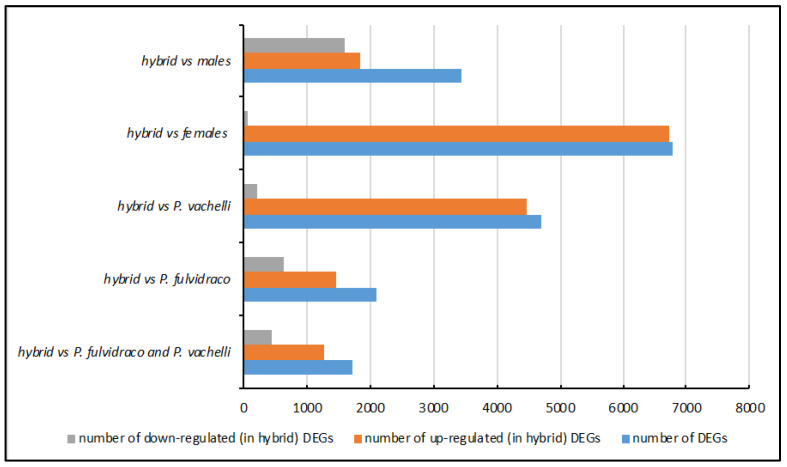
The numbers of DEGs in different groups.

**Figure 2 animals-14-01586-f002:**
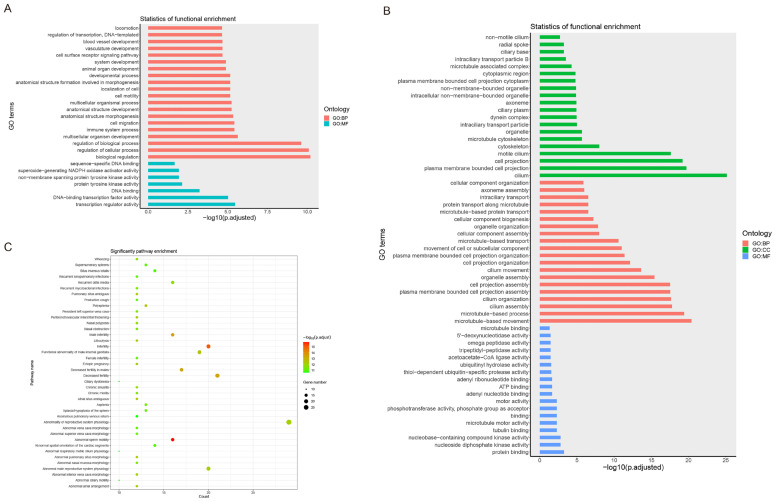
The enrichment analyses results for the DEGs between the hybrid and two pure yellow catfishes. (**A**) The GO enrichment result for the upregulated DEGs in the hybrid yellow catfish. (**B**) The GO enrichment result for the downregulated DEGs in the hybrid yellow catfish. (**C**) The HP enrichment result for the downregulated DEGs in the hybrid yellow catfish.

**Figure 3 animals-14-01586-f003:**
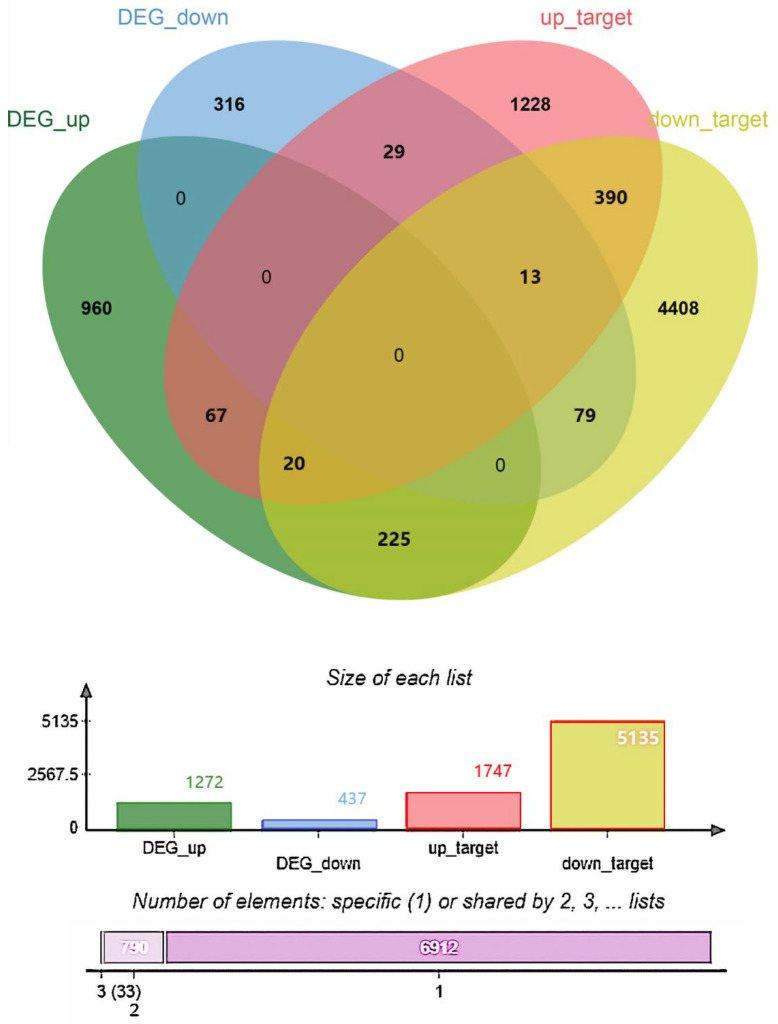
Correlation analysis of DEGs (DEG_up and DEG_down) and DEmiRNAs (down_target and up_target). The intersection genes between the upregulated DEGs (DEG_up) and the target genes of the downregulated DEmiRNAs (down_target) and between the downregulated DEGs (DEG_dowm) and the target genes of the upregulated DEmiRNAs (up_target). Cross regions represent the number of associated genes.

**Figure 4 animals-14-01586-f004:**
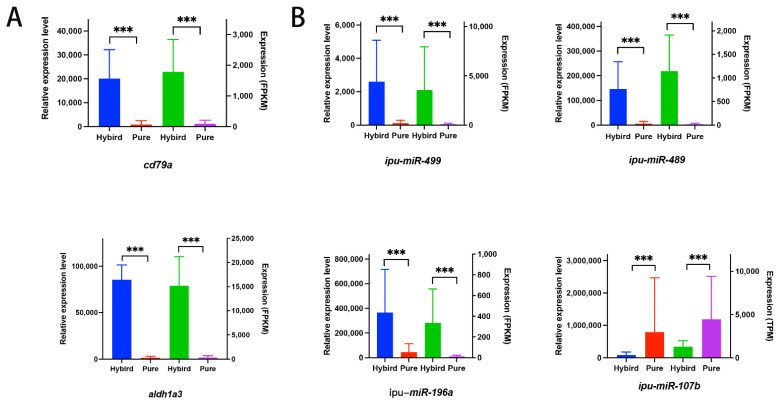
The results of RT-qPCR. (**A**) The results for two candidate DEGs. (**B**) The results for four candidate DEmiRNAs. The *p*-values were marked as “***” (≤0.001).

**Table 1 animals-14-01586-t001:** The numbers of DEGs in different groups.

Group	Number of DEGs	Number of Upregulated (in Hybrid) DEGs	Number of Downregulated (in Hybrid) DEGs
hybrid vs. *P. fulvidraco* and *P. vachelli*	1709	1272	437
hybrid vs. *P. fulvidraco*	2093	1467	626
hybrid vs. *P. vachelli*	4694	4476	218
hybrid vs. females	6788	6715	63
hybrid vs. males	3444	1844	1600

## Data Availability

Data are available on request from the corresponding author due to privacy/ethical restrictions.

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
