# Peer review of "Integrated miRNA and mRNA Sequencing Reveals the Sterility Mechanism in Hybrid Yellow Catfish Resulting from *Pelteobagrus fulvidraco* (♀) × *Pelteobagrus vachelli* (♂)"

_animals, 2024, doi:10.3390/ani14111586_

Round 1
Reviewer 1 Report
Comments and Suggestions for Authors
In the manuscript "Integrative miRNA and mRNA sequencing reveals sterility characteristics of Hybrid Yellow Catfish from Pelteobagrus fulvidraco (♀)×Pelteobagrus vachelli (♂)", the authors present some results regarding the use of a hybrid species, important for fisheries and aquaculture. Through miRNA sequencing, the authors identify some pathways and genes that may respond to certain DEmiRNAs responsible for infertility in the hybrid under investigation. However, the text provided by the authors contains serious errors in writing, agreement, syntax, coherence of ideas and methodological matters that require further explanation.
Line 55: Change “exist” by “exhibit”.
Line 71-74: “Although hybrid yellow catfish are almost sterile seriously limiting the development of the hybrid yellow catfish aquaculture industry, the hybrid sterility helps the conservation of genetic stock of yellow catfish.” Reference?
97-99: “Further studies are needed to explore the life processes and regulatory principles controlling sterility in hybrid yellow catfish.” What is it doing here “”? Same in line 101
Line 116: “advanced in fish evolution”. What does it mean, advanced in evolution? In evolutionary biology, we do not use the terminology "advanced" to describe a species in relation to its evolution. The term "advanced" can be interpreted in different ways and does not reflect the adaptation of a species.
Overall, the introduction is disconnected, extensive (unnecessary information), difficult to read and comprehend. There are many errors in agreement and argument connection. The introduction needs to be rewritten. Lacks flow.
Line 124: How was conducted fish maintenance?
Line 128: Properly cite companies name “...using 2-phenoxyethanol (Sigma) before...”
Line 133: “animal welfare and research laws”. Approval Number?
Line 123: Topic 2.1 requires rewriting for improved readability.
Lines 149-150: “After library construction, preliminary quantification was performed using Qubit 2.0, and the library was subsequently diluted.” Please, rephrase.
Line 188: How were GO, KEGG, and HP conducted? What is HP? Is it "HPO" (Human Phenotype Ontology)?
Line 212: Italicize gene names: “...genes (cd79a and aldh1a3) ...”
Line 226: Please review the literature to properly reformulate topic 2.7.
The materials and methods section requires a complete revision of the writing. As it is currently written, it is not possible to understand the logical and linear reasoning behind the experiments conducted by the authors and consequently review it.
Line 240: “Principal component analysis (PCA)”
Lines 240-241: “...so it was removed.” From subsequent analysis?
241-242: This is materials and methods: “...All yellow catfish specimens were over two years old, ensuring gonad maturation.”
Line 243-244: This sentence could be rephrased for more clarity “Totally obtained 79 million 150 bp paired end cleans reads, after removing the low-quality reads and adaptors.”
Lines 244-245: The authors provide one reference genome “all the clean reads were mapped to the reference genome”, but they used two different species, and one hybrid were used in the study. How did the authors handle it? Was only one reference genome was used for the three species? That’s why the mapping rate was low (66%)? Please, clarify.
How much of raw and clean data were obtained?
Lines 249-250: “Firstly, we focused on the DEGs between hybrid yellow catfish and two pure yellow catfishes (P. fulvidraco and P. vachelli).” Confuse. Please, clarify. Change “pure” for “wild”, or equivalent.
Instead of a table, the authors could use a Venn diagram to illustrate the DEGs in different groups, which would be much more didactic for readers.
Line 269: “...DEGs ((Table S2).” Remove one “(“.
It’s not necessary to use the number of each GO and HPO in the manuscript (GO:0048731), (HP:0012647), etc...
Enrichment analysis would benefit from an isolated topic. The authors use topic 3.2 for both DEGs and GO/HPO/KEGG
Figure 1 is in low resolution, hard to see the terms.
Lines 320-322: Rephase.
Line 324: “yellow catfish genome (ASM372403v1)”. Was only one genome was used again? Clarify.
Line 325: “16 teleost fishes as the background”. This information should be in the materials and methods.
Some supplemental tables could be converted into figures to enhance the quality of the authors' results presentation.
Lines 353-354: The sentence is confused. The authors could use: “In addition, we identified overlapping genes between the DEGs and the target genes of 353 DEmiRNAs across the hybrid and two wild groups”.
Reformulate the legend of Figure 2. The sentence is poorly written. The authors don't even mention the name of the graph. The same for figure 3.
Genes in figure 3 are not italicized.
Line 370: Italicize “genes (cd79a and aldh1a3)”.
Lines 383-384: Please, rewrite the sentence. The word hybrid was used 3 times in the same sentence.
Lines 387-388: Italicize genes “The genes involved in downregulation were identified as 387 CFAP70, TTC29, CCDC40, DNAAF4, SYCE3, STK36, SPI1, EMX2.”
399-401: “Therefore, we hypothesize that the significant downregulation of DEGs in hybrid yellow catfish is the primary cause of male infertility or reduced fertility.” This is an interesting hypothesis. Therefore, which DEGs the authors refer to? Clarify for readers.
Lines 407-409: Italicize genes “Representative genes involved in regulating these pathways include EMX2, SPI1, FOXD3, PDE1A, GCM2, ERG, OSR1, RBCK1, SMTNL2, MOS.”
Line 410: Clarify DEGs here “upregulation of DEGS”.
Line 447: Remove “speaking”.
Line 471: Genes spidr, cep85 and kcnn4 should be italicized. Please check the entire manuscript for correct gene and protein nomenclature.
Line 492: The word 'totally' is used frequently throughout the manuscript, often out of context and without regard to its intended meaning. Please review the entire manuscript for consistency in the use of this term.
Line 499: “totally” again.
Line 500: "Got" is an informal word, not suitable for a scientific manuscript.
Modify the Abstract following the suggestions above.
Comments on the Quality of English Language
Major review is necessary.
Reviewer 2 Report
Comments and Suggestions for Authors
Simple summary : no comments
Abstract: Abstract has too much results. An abstract means little bit of everything, may be rewrite it.
Introduction: paragraph ended in line 99 and began in line 100 did not matched well. From breeding process, its difficulties, the new paragraph just jumped into micro-RNA. There shall be few lines about possible genetic cause behind this problem, how to solve, some previous work and then come to micro-RNA.
Materials and methods: no comments
Results: no comments
Discussion: no comments
Conclusion: no comments
Round 2
Reviewer 1 Report
Comments and Suggestions for Authors
The authors have enhanced the manuscript's quality by doing the necessary revisions in the writing and improving clarity of the presented results.